# Solvothermally Synthesized Hierarchical Aggregates of Anatase TiO_2_ Nanoribbons/Nanosheets and Their Photocatalytic–Photocurrent Activities

**DOI:** 10.3390/nano13131940

**Published:** 2023-06-26

**Authors:** Kadhim Al-Attafi, Hamza A. Mezher, Ali Faraj Hammadi, Amar Al-Keisy, Sameh Hamzawy, Hamzeh Qutaish, Jung Ho Kim

**Affiliations:** 1Institute for Superconducting and Electronic Materials, Australian Institute for Innovative Materials (AIIM), University of Wollongong, North Wollongong, NSW 2500, Australia; hqmq581@uowmail.edu.au (H.Q.); jhk@uow.edu.au (J.H.K.); 2Department of Physics, College of Science, University of Kerbala, Karbala 56001, Iraq; hamza.a@uokerbala.edu.iq; 3Department of Mechanical Engineering, College of Engineering, Wasit University, Wasit 52001, Iraq; alifaraj@uowasit.edu.iq; 4Nanotechnology and Advanced Material Research Center, University of Technology-Iraq, Baghdad 10066, Iraq; Amar.H.Alkeisy@uotechnology.edu.iq; 5Intelligent Polymer Research Institute (IPRI), Australian Institute for Innovative Materials (AIIM), University of Wollongong, North Wollongong, NSW 2500, Australia; sameh.hamzawy@nriag.sci.eg; 6Solar Research Laboratory, Solar and Space Research Department, National Research Institute of Astronomy and Geophysics, Helwan 11421, Cairo, Egypt

**Keywords:** solvothermal synthesis, hierarchical anatase TiO_2_, photocatalytic activity, TiO_2_ nanoribbons/nanosheets, TiO_2_ nanoparticles

## Abstract

Hierarchical aggregates of anatase TiO_2_ nanoribbons/nanosheets (TiO_2_-NR) and anatase TiO_2_ nanoparticles (TiO_2_-NP) were produced through a one-step solvothermal reaction using acetic acid or ethanol and titanium isopropoxide as solvothermal reaction systems. The crystalline structure, crystalline phase, and morphologies of synthesized materials were characterized using several techniques. According to our findings, both TiO_2_-NR and TiO_2_-NP were found to have polycrystalline structures, with pure anatase phases. TiO_2_-NR has a three-dimensional hierarchical structure made up of aggregates of TiO_2_ nanoribbons/nanosheets, while TiO_2_-NP has a nanoparticulate structure. The photocatalytic and photocurrent activities for TiO_2_-NR and TiO_2_-NP were investigated and compared with the widely used commercial TiO_2_ (P25), which consists of anatase/rutile TiO_2_ nanoparticles, as a reference material. Our findings showed that TiO_2_-NR has higher photocatalytic and photocurrent performance than TiO_2_-NP, which are both, in turn, higher than those of P25. Our developed solvothermal method was shown to produce a pure anatase TiO_2_ phase for both synthesized structures, without using any surfactants or any other assisted templates. This developed solvothermal approach, and its anatase TiO_2_ nanostructure output, has promising potential for a wide range of energy harvesting applications, such as water pollution treatment and solar cells.

## 1. Introduction

Titanium dioxide (TiO_2_) is commonly employed in the fields of energy harvesting and energy storage, due to its unique properties [1,2]. It is widely used as a photocatalyst in various energy-related applications, including wastewater treatment, self-cleaning coatings, and solar energy conversion [2]. In order to achieve higher photocatalytic activity, materials with specific features should be utilized. These materials should have suitable band gap energy, high surface area, a well-defined crystal structure, high charge carrier mobility and lifetime, suitable band edge positions, and optimized catalyst loading [3]. Anatase TiO_2_ nanostructures, which aggregate different morphologies of nanomaterials into hierarchical structures, can result in the formation of complex structures, such as nano/microscale fibers, rods, spheres, and other shapes [4]. These structures can increase photocurrent properties as a result of their high surface area and charge transport, leading to enhanced photoelectrochemical performance [5].

In recent years, there has been considerable attention focused on hierarchical TiO_2_ nanostructures, due to their distinctive shape and outstanding photocatalytic capabilities when compared to their traditional counterparts [4]. Generally, hierarchical anatase TiO_2_ nanostructures exhibit superior photocatalytic performance compared to hierarchical rutile TiO_2_ nanostructures, primarily because of their unique surface characteristics and crystal structure [6]. Anatase TiO_2_ exhibits a greater surface area and an increased number of active sites than rutile TiO_2_, which can contribute to its improved photocatalytic activity. Additionally, the exposed (001) and (101) crystal facets of crystallized anatase TiO_2_ are more energetic than the (110) facets of rutile TiO_2_, resulting in increased photocatalytic activity [7]. Hierarchical anatase TiO_2_ nanostructures possess a unique crystal structure that promotes the enhanced mobility of charge carriers and a reduced rate of charge recombination, thereby improving the efficiency of photocatalytic reactions. Furthermore, these structures also have a higher surface area, thus providing more active sites for photocatalysis; meanwhile, improved charge transport can enhance the effective dissociation and transfer of light-induced charge carriers, leading to higher light absorption and charge separation [8]. Consequently, hierarchical anatase TiO_2_ nanostructures offer better reproducibility and scalability, making them more appealing for practical applications than hierarchical rutile TiO_2_ [9].

Polycrystalline anatase TiO_2_ nanostructures have multiple crystal domains with different orientations and crystal facets, which can lead to a lower surface area and lower crystallinity. However, the presence of multiple crystal domains can lead to a higher density of defect states, which, in turn, can act as trapping centers through which photogenerated charge carriers can be separated and transferred more efficiently, and thus, increasing the photocatalytic activity [10]. Additionally, the reactivity of single or polycrystalline TiO_2_ nanostructures towards different types of photoreactions can also differ depending on their crystal structure; for example, the single-crystal TiO_2_ is suitable for photocatalysis and surface science studies, while the polycrystalline TiO_2_ is commonly used due to its availability, cost-effectiveness, and ease of production [11].

The crystal facets of hierarchical aggregates of anatase TiO_2_ can significantly affect their photocatalytic activity [12]. Anatase TiO_2_ has several crystal facets, including (001), (100), and (101) planes, which exhibit different surface energies and reactivity toward different types of photoreactions [13]. Hierarchical aggregates of anatase TiO_2_ can exhibit multiple crystal facets, which can influence the photocatalytic activity by affecting the adsorption, generation, separation of charge carriers, and surface reactivity. The photocatalytic performance of hierarchical aggregates of anatase TiO_2_ is possible to tune by controlling the relative exposure of different crystal facets [14]; for example, (001) facets are more reactive towards oxygen species and are responsible for the generation of hydroxyl radicals, while (101) facets are active toward the reduction in oxygen species and can facilitate the production of hydrogen gas from water-splitting reactions [14,15].

Hierarchical aggregation is the process of combining multiple individual nanomaterials into a larger structure with a higher level of order [4]. This can be achieved through many physical and chemical techniques, such as self-assembly, solution-based gel, electrodeposition, hydrothermal or solvothermal synthesis, and template-assisted synthesis [16,17,18,19,20]. Solvothermal synthesis is the commonly employed technique with which to produce hierarchical aggregates of anatase TiO_2_ for photocatalytic applications, due to the economic feasibility of this method and the possibility of preparing suitable shapes and sizes of nanomaterials [21,22,23].

Researchers have recently focused on TiO_2_ nanostructures exhibiting various shapes and nanocomposites based on titania, as they possess a wide range of distinct physicochemical properties [24]. Nanostructures of titanium dioxide (TiO_2_) with one-dimensional (1D) characteristics, including nanorods, nanotubes, and elongated cylindrical structures, have been observed to display superior photocatalytic performance when compared to P25 particles. This enhanced activity can be attributed to the improved separation of the electron–hole (e−/h+) pairs and decreased occurrence of charge recombination [25]. Despite the widespread use of TiO_2_ photocatalysts, they still have some limitations that affect their photocatalytic activity. Some of these limitations include their poor absorption of sunlight and rapid recombination of photogenerated electrons/holes, which continue to hinder their widespread application [26]. However, research has shown that the porous microstructure of TiO_2_ is correlated with improved photocatalytic activity, specifically TiO_2_ with a hierarchical structure and multiple levels of nanostructures that are of a particular interest. It has been reported that the hierarchical three-dimensional (3D) TiO_2_ nanospheres, which contain one-dimensional (1D) nanorods, exhibit exceptional photocatalytic performance due to their electron transfer capabilities [27]. The synthetic protocols for hierarchical TiO_2_ materials have become more advanced, allowing researchers to manipulate and regulate characteristics such as structural arrangement, particle dimensions, shape, and surface characteristics [4]. Designing photocatalysts that possess a hierarchical structure at both micrometer and nanometer dimensions can effectively address numerous obstacles associated with the thermodynamic and kinetic characteristics of a photocatalyst [4]. Hierarchical structures with connected porous networks can help reactants move toward the active sites on the walls of the pores. This allows for better diffusion and enhances various properties, such as improved absorption of light, faster movement of molecules, larger surface area, and more active sites [28]. Nevertheless, achieving a suitable equilibrium among the hydrolysis rate of the titanium precursor, the growth rate of titanium dioxide, and the desired orientation makes it a difficult task during hydrothermal/solvothermal conditions [29].

The precursor is generally prepared by dissolving a suitable Ti precursor in a suitable solvent and then subjecting it to a solvothermal reaction [30]. This process leads to the growth and self-assembly of larger, hierarchical aggregates, which can form complex structures such as hollow spheres, flower-like structures, and dendritic structures. Those structures were found to have enhanced photocatalytic performance due to their larger surface area, enhanced ability to transfer charges, and improved capacity to absorb light [22].

Various methods can be used to synthesize hierarchical TiO_2_, including those involving surfactants or templates, and those without either of them. The choice of synthesis method has a profound impact on the produced TiO_2_ material’s morphology, crystallinity, and surface properties. Extensive research explored the advantages and disadvantages of both surfactant/template-assisted and surfactant/template-free approaches for the synthesis of hierarchical TiO_2_ [4,31,32].

Surfactants and templates were found to be essential in directing the growth of TiO_2_ nanoparticles and facilitating their self-assembly into hierarchical structures. Surfactants, such as Cetrimonium bromide (CTAB) or Polyvinylpyrrolidone (PVP), assist in stabilizing the nanoparticles and controlling their size and shape. Templates provide a framework for hierarchical assembly, to help in producing specific morphologies such as nanorods, nanotubes, or mesoporous structures [33].

Numerous studies have successfully synthesized hierarchical TiO_2_ using surfactant or template methods [34]. For example, in 2017, Bhat et al. utilized a template-assisted solvothermal method in order to synthesize and control mesoporous hierarchical TiO_2_ spheres [35]. These spheres exhibited a high surface area and enhanced photocatalytic activity. In the same year, Hu et al. employed a surfactant-templated hydrothermal approach in order to synthesize TiO_2_ nanoparticles and nanowires that showed improved photocatalytic performance [36].

On the other hand, the surfactant/template-free synthesis of hierarchical TiO_2_ has been considered an alternative method with distinct advantages [37,38]. This approach eliminates the need for additional purification steps and overcomes potential issues associated with surfactant or template residues. Moreover, it provides a significant control over the morphology and surface properties of the resulting TiO_2_ structures [38].

Recent studies have reported the successful synthesis of hierarchical TiO_2_ without surfactants or templates. In 2013, Chen et al. developed self-assembled ultrathin TiO_2_ hierarchical nanostructures using a surfactant-free sol–gel method. These hierarchical nanostructures additionally demonstrated improved photocatalytic performance compared to conventional TiO_2_ materials [38]. Earlier this year, Yu et al. achieved surfactant-free hydrothermal synthesis of hierarchical TiO_2_ nanosheets with exposed (001) facets [39]. These nanosheets exhibited enhanced photocatalytic activity, which was attributed to their unique morphology and increased surface area.

Various synthetic routes were explored for producing hierarchical TiO_2_ nanostructures exhibiting both 2D and 3D shapes. Nowadays, researchers are actively investigating a variety of methods for synthesizing hierarchical TiO_2_ nanostructures with a wide range of porosity and controlled morphologies. The control of particle size and crystallographic nanostructure orientation is crucial to achieving high performance and reusability [40].

Recently, Vidyasagar et al. and Wang et al. prepared hierarchical meso-macroporous nanoflowers with mesoporous and macroporous structures via a template-assisted sol–gel method, and demonstrated their excellent performance for the degradation of phenol and methylene blue under visible and ultraviolet lights, respectively [41,42]. Moreover, Zhu et al. also synthesized hierarchical mesoporous TiO_2_ microspheres with macroporous structures via solvothermal synthesis, with a facile formation mechanism and enhanced photocatalysis [43]. Hongwei Bai et al. presented the synthesis and characterization of 3D dendritic TiO_2_ nanospheres with extremely long 1D nanoribbons/wires for the simultaneous purification of water using photocatalytic membranes. These studies highlight the potential of these structures for efficient photocatalysis, offering improved performance due to their unique morphology and enhanced charge separation, as well as their transport properties. Further research is still ongoing to develop advanced hierarchical structures with advanced photocatalysis capabilities [29].

The synthesis of hierarchical TiO_2_ with/without surfactants or templates presents distinct advantages and disadvantages. Surfactant/template-assisted methods offer precise morphology control, but may introduce impurities and interfere with photocatalytic properties. Meanwhile, surfactant/template-free methods provide simplicity, cost-effectiveness, and enhanced photocatalytic activity, but they may also have limitations in morphology control. Researchers are actively exploring novel synthesis strategies in order to customize hierarchical TiO_2_ structures for diverse applications, aiming to strike a balance between the advantages offered by both approaches.

In this study, we synthesized the hierarchical structures of anatase TiO_2_ nanoribbons/nanosheets and anatase TiO_2_ nanoparticles using template/surfactant-free solvothermal reaction. Their photocatalytic activities and photocurrents were also compared with those of the commercial (P25) TiO_2_. The anatase TiO_2_ nanoribbons/nanosheets showed unique morphology, with elongated shapes and large surface area. The hierarchical anatase TiO_2_ nanoribbons or nanosheets are favorable materials for photocatalytic applications, due to their large surface area, high reactivity, and excellent photocatalytic properties. Their photocatalytic and photocurrent performances were found to be improved by aggregating them into hierarchical structures, in comparison to the commercial P25, which is widely used as a benchmark in photocatalysis research as an ideal standard material for measurements.

## 2. Experimentation, Materials and Characterizations

TiO_2_-NR and TiO_2_-NP were synthesized using a solvothermal method. A precursor of titanium isopropoxide (TTIP, supplied by Sigma, Macquarie Park, NSW, Australia) was used, while acetic acid (Sigma, Macquarie Park, NSW, Australia) and absolute ethanol (Sigma, Macquarie Park, NSW, Australia) were used as solvents. TTIP (1.5 mL) was slowly added to ethanol or acetic acid, while vigorously stirring at room temperature for one hour. From this, a white solution was obtained, which was then subsequently moved into a (45 mL) stainless steel autoclave lined with Teflon. (Manufactured by Parr Instrument Company, Moline, IL, USA). After maintaining the autoclave at a temperature of 180 °C for 9 h, a white precipitate formed upon cooling it to ambient temperature. The precipitate was rinsed twice using a mixture of ethanol and distilled water, and subsequently dried overnight at a temperature of 90 °C. Finally, TiO_2_-NR and TiO_2_-NP powders were sintered at 450 °C at a rate of 1 °C per minute in the presence of air for 3 h. The commercial Degussa TiO_2_ (P25) (Sigma, Macquarie Park, NSW, Australia), was used as received.

The synthesized materials were characterized using various techniques. The crystalline structure of materials was analyzed using an X-ray diffractometer (manufactured with GBC Scientific Equipment LLC, Hampshire, IL, USA). The X-ray instrument was set at scan range = 20°–80°, voltage = 40 kV, current = 30 mA, and a wavelength of Cu Kα radiation = 1.54 Å. The physical characteristics of the samples, including their morphology, internal structure, and element composition, were analyzed using field-emission scanning electron microscopy (FE-SEM) with a JEOL JSM-7500 instrument (Tokyo, Japan), and transmittance electron microscopy (TEM) with a JEOL JEM-6500F instrument (Tokyo, Japan). The Brunauer–Emmet–Teller (BET) surface area, as well as the porosity and pore volume obtained with BHJ (Barrett, Joyner, and Halenda), was determined by collecting the data on Microtrac Belsorp mini equipment (Osaka, Japan). The photocatalytic activity experiments were conducted by dispersing TiO_2_-NR, TiO_2_-NP, and P25 in a water-based solution containing Rhodamine B dye (purchased from Sigma, 95% purity). The catalyst was added at a concentration of around 20 mg per 20 mL of dye solution, with a concentration of 25 µM. The measurements were conducted using simulated sunlight illumination with the Oriel LCS-100 at an intensity of 100 mW/cm^2^. The photocurrents of TiO_2_-NR, TiO_2_-NP, and P25 were measured using 1 V applied voltage, 300 W Xenon light, Na_2_SO_4_ electrolyte, 50 s on–off time, and 1 × 1 cm^2^ thin film area. Measurements of the optical energy gaps of TiO_2_-NR and TiO_2_-NP were performed utilizing a combination of a Shimadzu UV-3600 spectrophotometer and an attached integrating sphere (ISR-3100) (Tokyo, Japan).

## 3. Results and Discussions

### 3.1. Proposed Synthesis Mechanism

The choice of solvent can significantly affect the formation of hierarchical aggregates of anatase TiO_2_ nanoribbons/nanosheets. Ethanol and acetic acid are commonly employed as solvents in solvothermal synthesis due to their capability to dissolve Ti precursor species and effectively control the growth and morphology of the resulting nanostructures [44,45]. Ethanol is a polar solvent with a low boiling point, while acetic acid is a weak acid with a higher boiling point. Acetic acid plays a significant role in facilitating the hydrolysis and condensation of titanium precursor compounds. Additionally, it can also regulate the acidity or alkalinity of the solution involved in the reaction [46,47,48]. In order to achieve the desired morphology and properties for enhanced photocatalytic applications, meticulous selections and optimizations of the solvents and the reaction conditions should be considered [49,50].

In order to understand the mechanism, the synthesis of TiO_2_-NR and TiO_2_-NP was performed without using surfactants. After allowing the solvothermal reaction to proceed for 9 h at 180 °C, amorphous Ti chain precursors were formed. The porosity of the interconnected groups of chains was controlled via the esterification of the organic acid with alcohol in the reaction system, which is released as TTIP dissociates [13]. In the early phases of the process, the amorphous titanium chains are surrounded by water-repellent acetate groups, and the outer surfaces of these chains attach to esters within the chemical reaction setup. The carboxyl groups in the solvent precursor are expected to coordinate with Ti atoms, forming bidentate complexes through the interaction with an organic ligand present in the titanium-containing precursor [51]. The proposed formation process of the TiO_2_-NR includes the process of TTIP acidolysis, facilitated by acetic acid as a catalyst to form Ti chains. Afterward, the esters facilitate interactions that promote the proximity of the chains and allow their organization into regular repeating patterns. In the previous study conducted by our group, the initial stages of formation for amorphous Ti chains were explained when synthesizing anatase single-crystal TiO_2_ using a similar titanium alkoxide, specifically titanium butoxide, along with acetic acid [13]. Subsequently, subjecting anatase TiO_2_ to calcination at a temperature of 450 °C for 3 h in air leads to the creation of networks comprising Ti-O-Ti and O-Ti-O bonds. Finally, the calcination step leads to the formation of TiO_2_-NR or TiO_2_-NP, where esters serve as an implicitly established self-template during the formation process [45]. It is proposed that the resulting phases that emerged from the initial interaction between the TTIP and acetic acid or ethanol, along with the interplay between the Ti source and solvent, play a crucial role in the formation of the porous structure and morphology of the TiO_2_-NR and TiO_2_-NP. The hierarchical nanostructure of TiO_2_-NR was successfully achieved through a straightforward and precisely controlled solvothermal synthesis method known as “soft”, based on acetic acid as a “structure-directing effect”.

### 3.2. Structural and Optical Characterizations

The crystalline phases of TiO_2_-NR and TiO_2_-NP were confirmed via X-ray diffraction (XRD) data. The diffraction patterns observed represent the crystal planes of anatase TiO_2_, which was further confirmed in their corresponding selected area electron diffraction patterns (SAED). This implies the successful formation of the anatase phase, with a highly crystalline structure for both TiO_2_-NR and TiO_2_-NP, as shown in Figure 1a,b, (XRD data card, JSPD.21-1272).

As shown in Figure 2a, the scanning electron microscope (SEM) image of TiO_2_-NR revealed the formation of three-dimensional interconnected structures involving nanoribbon/nanosheet structures, suggesting the hierarchical aggregation of the synthesized material. However, the SEM image of TiO_2_-NP (Figure 2b) revealed the formation of rounded TiO_2_ nanoparticles. The insets in Figure 2a,b represent the energy dispersive X-ray diffraction (EDX) spectra that confirmed the pure chemical composition of both TiO_2_-NR and TiO_2_-NP (Cu and C peaks refer to the elements of the grid used in the SEM measurements). The inset histograms quantified the average particle size distributions, which were measured as 1.7 ± 0.3 µm and 18 ± 5 nm for TiO_2_-NR and TiO_2_-NP, respectively.

Furthermore, the internal structure (i.e., the size and shape of the nanoparticles) for TiO_2_-NP and TiO_2_-NR were investigated using TEM and HRTEM images, and the results were presented in Figure 3a–f. Figure 3a,b show that TiO_2_-NP contained nanoparticles with an average size of 18 ± 5 nm, and the shape of these particles was primarily determined by the extent to which the octahedral structure is shortened during the solvothermal process [52,53]. Moreover, Figure 3c–f exhibited two distinct morphologies within the spherical TiO_2_-NR, namely nanoribbons that formed aggregated spindle nanoparticles, and nanoparticles that formed aggregated nanosheets, with a size range of 18 ± 5 nm in size. The insets in Figure 3a,e,f showed the shape morphologies of aggregated nanoparticles in TiO_2_-NP and TiO_2_-NR. The crystal structures of the primary TiO_2_ nanoparticles, which aggregated to form TiO_2_-NP and TiO_2_-NR, were further confirmed with high-resolution TEM (HRTEM) images in Figure 3b,e,f. This confirmation was achieved by analyzing the interplanar distances and their corresponding Miller indices. HRTEM findings further confirmed the XRD and SAED analyses in Figure 1a,b.

It is expected that TiO_2_-NR, with both truncated and spindle-shaped nanoparticles, will have a higher number of exposed energetic (001) facets compared to TiO_2_-NP. These differences in morphology between the synthesized materials were mainly attributed to the differences in the synthesis reaction solvents.

XRD, SEM, and TEM characterizations have clearly shown that TiO_2_-NR has a hierarchical structure made up of tiny anatase TiO_2_ nanoribbons/nanosheets, which form a highly interconnected mesoporous structure compared to TiO_2_-NP. Additionally, the BET analysis obtained from N_2_ adsorption/desorption isothermal data (illustrated in Figure 4a and Table 1) has also confirmed the formation of a highly mesoporous structure in TiO_2_-NR compared to TiO_2_-NP and P25.

The structure of TiO_2_-NR has a greater surface area and cumulative pore volume, due to the presence of a higher amount of condensed nitrogen in the large voids and pores of TiO_2_-NR. Furthermore, the average pore size distributions obtained with the BJH analysis (inset in Figure 4a) have revealed that the internal pore size of TiO_2_-NR is slightly smaller than that of TiO_2_-NP due to the slightly smaller size of their aggregated primary nanoparticles. The voids among the hierarchical structures of TiO_2_-NR are expected to be approximately submicrosized; however, this is not observable due to the measurement limit of the BET equipment. The BET calculations shown in Table 1 suggest that TiO_2_-NR will be expected to accommodate a greater amount of dye, due to higher surface area and porosity compared to both TiO_2_-NP and P25. This indicates enhanced surface reactivity, as well as the potential for improved photocatalytic activity. Moreover, the unique morphology of TiO_2_-NR, characterized by highly interconnected nanoribbons/nanosheets, forms a hierarchical structure. This structure serves as an efficient pathway for charge diffusion, both within the material’s internal pores and along its external surface. Consequently, the presence of these pores is expected to enhance the photocurrent.

Figure 4a and Table 1 show the higher surface area, cumulative pore volume, porosity, and roughness factor of TiO_2_-NR in comparison to TiO_2_-NP and P25. On the other hand, the average pore size was observed to be similar to those of both TiO_2_-NR and TiO_2_-NP. This indicates the increased surface reactivity of TiO_2_-NR compared to TiO_2_-NP and P25. It should be noted that performing Hg intrusion in addition to N_2_ adsorption/desorption isotherms is recommended in order to detect larger pore sizes.

Figure 4b showed the calculated energy band gaps (E_g_) of TiO_2_-NR and TiO_2_-NP using the Tauc plots. The E_g_ values of TiO_2_-NR and TiO_2_-NP were found to be 3.23 eV and 3.21 eV, respectively, which are almost the same. These values confirm the ultraviolet absorption region of anatase TiO_2_, which can serve as a photocatalyst.

### 3.3. Photocatalytic and Photocurrent Characterizations

To assess the photocatalytic activity of TiO_2_-NR and TiO_2_-NP, photocatalytic degradation experiments were conducted, using a model organic pollutant, such as organic Rhodamine B, as a standard dye compound. The photocatalytic performances of TiO_2_-NR and TiO_2_-NP were determined by measuring the degradation rate, or measuring the percentage of Rhodamine B degradation over a certain time, and then comparing it against that of the standard material, TiO_2_ (P25). This is typically measured under ultraviolet and visible lights. Photocurrent characterizations were also performed, in order to evaluate the charge transport properties of TiO_2_-NR and TiO_2_-NP using photocurrent transient measurements. As shown in Figure 5a–d, TiO_2_-NR and TiO_2_-NP exhibited higher photocatalytic performance than that of the standard, P25; however, TiO_2_-NR still showed better performance compared to TiO_2_-NP. The unique morphology of TiO_2_-NR, in addition to its high surface area and hierarchical shape (i.e., 3D hierarchically aggregated 1D nanoribbons/2D nanosheets structure) provided increased light absorption and efficient charge transfer. This results in enhanced photocatalytic performance. The standard P25 exhibited the lowest photocatalytic/photocurrent activities, due to its lowest surface area, cumulative pore volume, porosity, roughness factor, and pore size. These photocatalytic measurements were performed on P25 under similar conditions as a catalyst reference material, and were recorded and listed in Table 1.

## 4. Conclusions

Our developed solvothermal synthesis technique, utilizing TTIP as a precursor and acetic acid or ethanol as solvents, without using templates or surfactants, was discovered to generate a hierarchical aggregation of anatase TiO_2_ nanoribbons/nanosheets (TiO_2_-NR) and anatase TiO_2_ nanoparticles (TiO_2_-NP). Remarkably, these synthesized materials exhibited superior photocatalytic and photocurrent performances, compared to the commercial TiO_2_ (P25). Additionally, the synthesis process allowed for the production of two anatase TiO_2_ nanostructures with different sizes, shapes, and morphologies, which, in turn, optimized photocatalytic performance. The enhanced photocatalytic and photocurrent activities of TiO_2_-NR were found to be attributed to multiple factors. Firstly, the unique morphology of TiO_2_-NR provides a larger surface area, promoting improved light absorption and increasing the number of active sites (such as the exposed facets 001 and 101) for efficient photocatalytic reactions. Secondly, the hierarchical aggregation of TiO_2_-NR results in interconnected networks, facilitating the efficient transport of photo-generated charges and minimizing recombination losses. On the other hand, TiO_2_-NP exhibited lower photocatalytic and photocurrent activities because of its reduced surface area and increased recombination losses. These effects can be attributed to the spherical nanoparticulate shape of TiO_2_-NP and the increased boundaries among the aggregated nanoparticles. This further substantiated the superior photocatalytic performance of TiO_2_-NR. The photocurrent measurements of the materials also indicated that the anatase TiO_2_-NR and TiO_2_-NP still exhibited improved charge transport properties compared to the standard P25, suggesting their potential for many energy applications, such as solar energy harvesting, water purification, and other energy-related applications.

## Figures and Tables

**Figure 1 nanomaterials-13-01940-f001:**
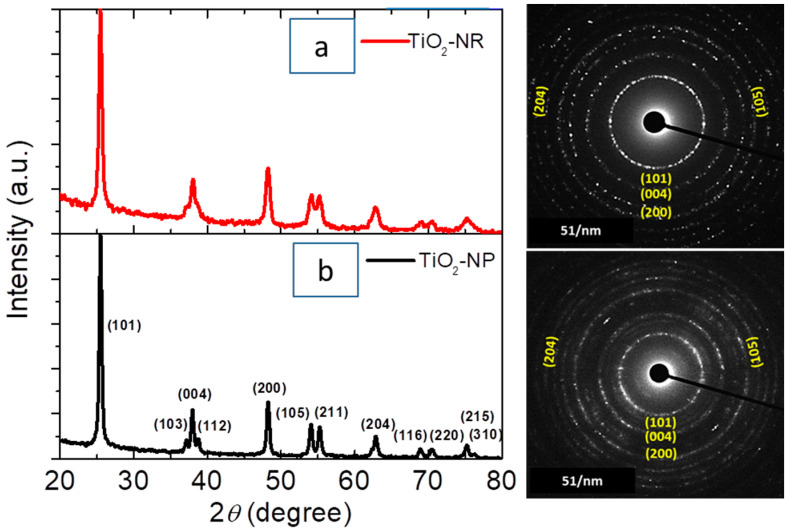
XRD patterns and their corresponding SAED patterns for (**a**) TiO_2_-NR and (**b**) TiO_2_-NP.

**Figure 2 nanomaterials-13-01940-f002:**
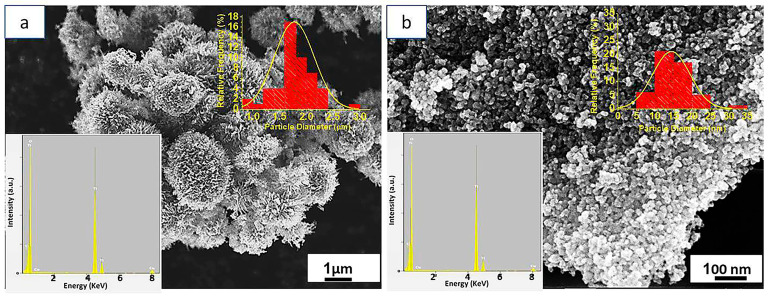
SEM and inset EDX images of (**a**) TiO_2_-NR and (**b**) TiO_2_-NP.

**Figure 3 nanomaterials-13-01940-f003:**
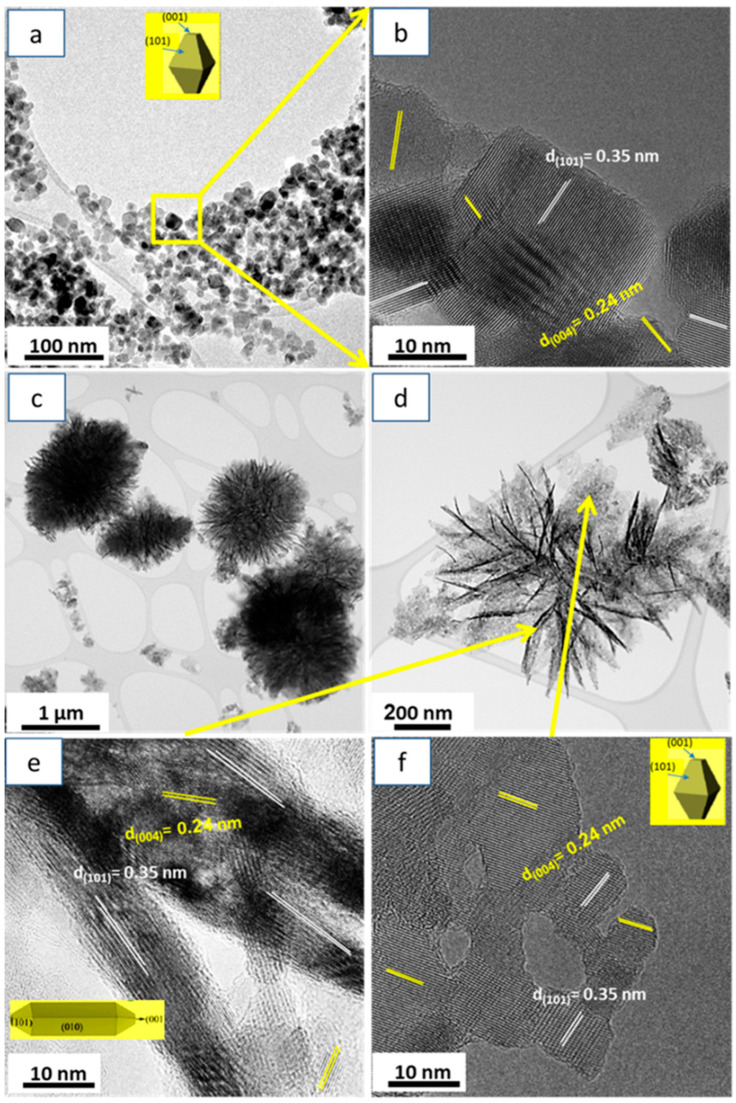
TEM and HRTEM images of (**a**,**b**) TiO_2_-NP and (**c**–**f**) TiO_2_-NR.

**Figure 4 nanomaterials-13-01940-f004:**
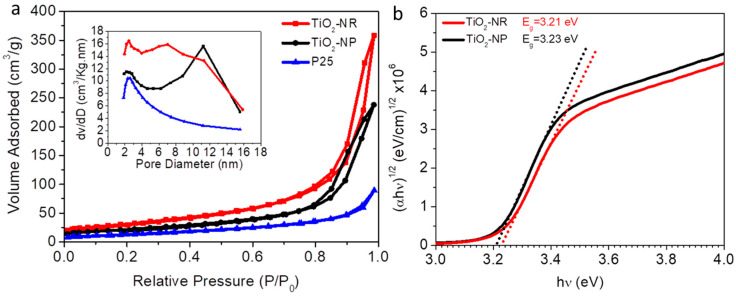
(**a**) BET surface area and the inset BJH pore size measurements of TiO_2_-NR, TiO_2_-NP and P25 (**b**) Tauc plots of energy gap measurements of TiO_2_-NR and TiO_2_-NP.

**Figure 5 nanomaterials-13-01940-f005:**
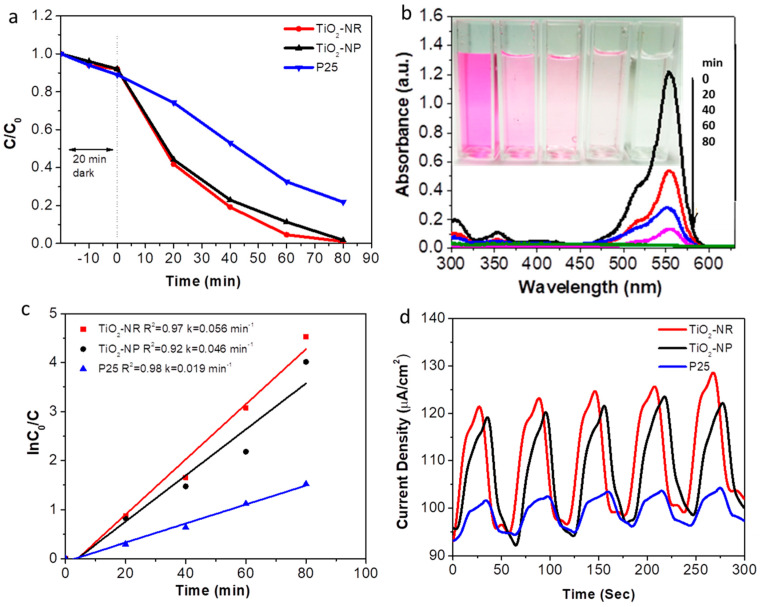
(**a**–**c**) The photocatalytic measurements and (**d**) photocurrent measurements of TiO_2_-NR, TiO_2_-NP, and P25.

**Table 1 nanomaterials-13-01940-t001:** The porosity (P), surface area (S_a_), pore size (P_d_), and roughness factor (R_f_), of TiO_2_-NR, TiO_2_-NP, and P25. Porosity was calculated using P = P_V_/(1/ρ + P_V_), where (P_V_) is the cumulative pore volume, and ρ is the density value of TiO_2_ (0.257 cm^3^/g) [54].

Material	Pv (cm^3^/g)	P (%)	S_a_ (m^2^/g)	R_f_ (µ/m)	P_d_ (nm)
TiO_2_-NP	0.18	41	116	183	7.0
TiO_2_-NR	0.24	48	80	234	6.5
P25	0.07	22	50	150	5.0

## Data Availability

Not applicable.

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
