# Peer review of "Solvothermally Synthesized Hierarchical Aggregates of Anatase TiO2 Nanoribbons/Nanosheets and Their Photocatalytic–Photocurrent Activities"

_nanomaterials, 2023, doi:10.3390/nano13131940_

Round 1

Reviewer 1 Report

It is a good work, and may be published after the following revisions:

1. The photocatalytic properties of the as-synthesized TiO2 had better be studied using colorless pollutants.

2. The dark adsorption results should be included in Figure 5a.

3. The photocatalytic stability had better be studied.

good

Author Response

Dear reviewer,

Firstly, we would like to thank the referees and editors for their time and efforts in the consideration of this manuscript. We have read and considered their responses and believe we have addressed their concerns point-by-point as below:

Comments and Suggestions for Authors

It is a good work, and may be published after the following revisions:

  1. The photocatalytic properties of the as-synthesized TiO2 had better be studied using colorless pollutants.

Re.: In this study, we focused on comparing our products with commercial TiO2 (P25) and we observed that TiO2-NR and TiO2-NP had higher photocatalytic activity when compared with P25. This was determined through an investigation of the photodegradation with a commonly available standard organic Rhodamine B dye. However, additional investigation, for example, photodegradation for colorless contaminants, photo-oxidation drugs, and reduce heavy metals in wastewater; it will be taken into consideration in the subsequent or future studies.

  1. The dark adsorption results should be included in Figure 5a.

Re.: The dark adsorption results were already included in Figure 5 a, but they weren’t scaled in it.

Figure 5 a has now been edited and rescaled.

  1. The photocatalytic stability had better be studied.

Re.: Since it is well known that the stability of TiO2 is extremely high. It has been extensively studied and proven to be photocatalytically stable in previous publications. We believe that it is not worth; conducting the experiment of photocatalytic stability for TiO2 as it will not provide any novelty to the manuscript.

Best regards,

Reviewer 2 Report

The manuscript reports on anatase TiO2 nanoribbons/nanosheets (TiO2-NR) and anatase TiO2 nanoparticles (TiO2-NP) prepared by a one-step solvothermal synthesis and exhibiting higher performance in photocatalytic degradation of Rhodamine B and photocurrent. This is an interesting and noteworthy piece of work. However, the authors do not explain their view clearly enough and lacking relevant characterization tests to support their view. Before considering it to be published, some suggestions are provided as follows:

1.      In "3.1. Synthesis mechanism" of the manuscript, the explanation of the mechanism of synthesis of TiO2-NR and TiO2-NR without surfactant lacks a basis in the literature and the reviewer suggests citing references.

2.      The authors have measured the specific surface area, pore volume and pore size of different catalysts and compared them with P25, but the experimental data for P25 is missing in Figure 4a. The reviewer suggested that the author supplement the data to make it more complete.

3.      In the manuscript, the performance tests on the TiO2-NR and TiO2-NP catalysts for the photocatalytic degradation of Rhodamine B (Figure 5a-c) do not reflect whether the catalysts reached adsorption equilibrium for the substrate at the beginning of the test, and whether this was one of the reasons for the fastest degradation rates in the first 20 minutes of the test.

4.      In the manuscript, the authors' evaluation of the photocatalytic performance of the P25, TiO2-NR and TiO2-NR catalysts lacks catalyst stability tests to reflect the fact that the modified TiO2 has higher performance while its stability is not compromised.

5.      There are some formatting errors in the manuscript. In the "1. Introduction" section (line 81), the references are cited in inconsistent case, e.g. "[13]". In the section "2. Experimental, Materials and Characterisations", there is a formatting error such as "TiO2", which should be changed to "TiO2".

Author Response

Dear reviewer, 

Firstly, we would like to thank the referees and editors for their time and efforts in the consideration of this manuscript. We have read and considered their responses and believe we have addressed their concerns point-by-point as below:

Comments and Suggestions for Authors

The manuscript reports on anatase TiO2 nanoribbons/nanosheets (TiO2-NR) and anatase TiO2 nanoparticles (TiO2-NP) prepared by a one-step solvothermal synthesis and exhibiting higher performance in photocatalytic degradation of Rhodamine B and photocurrent. This is an interesting and noteworthy piece of work. However, the authors do not explain their view clearly enough and lacking relevant characterization tests to support their view. Before considering it to be published, some suggestions are provided as follows:

  1. In "3.1. Synthesis mechanism" of the manuscript, the explanation of the mechanism of synthesis of TiO2-NR and TiO2-NR without surfactant lacks a basis in the literature and the reviewer suggests citing references.

Re.: Further literature research works about the synthesis of hierarchical TiO2 nanostructures with/without using surfactants and their advantages/disadvantages have been added to the introduction. More references and explanations have been added to the synthesis mechanism section. 

  1. The authors have measured the specific surface area, pore volume, and pore size of different catalysts and compared them with P25, but the experimental data for P25 is missing in Figure 4a. The reviewer suggested that the author supplement the data to make it more complete.

Re.: The experimental data for P25 has been added to Figure 4a and the inset.

  1. In the manuscript, the performance tests on the TiO2-NR and TiO2-NP catalysts for the photocatalytic degradation of Rhodamine B (Figure 5a-c) do not reflect whether the catalysts reached adsorption equilibrium for the substrate at the beginning of the test and whether this was one of the reasons for the fastest degradation rates in the first 20 minutes of the test.

Re.: The dark adsorption results are included in Figure 5 a, but they weren’t scaled in it.

Figure 5 a has now been edited and rescaled with the dark adsorption data.

  1. In the manuscript, the author's evaluation of the photocatalytic performance of the P25, TiO2-NR, and TiO2-NR catalysts lacks catalyst stability tests to reflect the fact that the modified TiO2 has higher performance while its stability is not compromised.

Re.: Since it is well known that the stability of TiO2 is extremely high. It has been extensively studied and proven to be photocatalytically stable in previous publications. We believe that it is not worth; conducting the experiment of photocatalytic stability for TiO2 as it will not provide any novelty to the manuscript.

  1. There are some formatting errors in the manuscript. In the "1. Introduction" section (line 81), the references are cited in inconsistent case, e.g. "[13]". In the section "2. Experimental, Materials and Characterisations", there is a formatting error such as "TiO2", which should be changed to "TiO2".

Re.: The formatting errors have been corrected and the references rearranged using the MDPI ACS reference style.

Best regards,

Authors,

Reviewer 3 Report

The paper reports on photocatalytically highly active hierarchical TiO2 nanoribbons/nanosheets prepared solvothermally. As test reaction only the – well known, but also very unspecific – methylene blue degradation is used.
Many of the claims given in the paper are not well proven and documented. Moreover, there are a lot of spelling and grammar errors in the text.

In total, I cannot recommend publication of the paper, since a number of facts are not well presented – and especially scientific explanations are missing.

Nevertheless, the paper is widely well written and can be published. However, the authors should consider the following points in a revision of the paper:

1. Chapter 3.1: The synthesis mechanism remains unclear. Especially I do not understand for what reasons either nanoribbons or nanoparticles form? What initiates the formation of the nanoribbons? Is there any proof for the proposed “regular repeating structures” (line 207)? How can the ratio of nanoribbons to nanoparticles formed be controlled? The last sentence of the paragraph claims that by-products (line 211) play a role – what kind of by-products are these and do the authors have any proof for their formation?

2. In the paragraph starting below Fig. 2, i.e., lines 234 – 246 the authors write several times “Fig. 6”, but they mean “Fig. 3”.

3. Line 240: Where are the “700 nanoribbons…” coming from? This sentence is not clear to me – please explain with more details.
In line 264 it is said that the nanoribbons might be 700 nm in size – what might define that size, i.e., what limits the interaction of the nanospindles and nanosheets to that size? The mechanism of the aggregation remains fully unclear.

4. Fig. 4: Since at p/p0 close to 1 the adsorption is still increasing strongly, it must be assumed that also pores with diameters larger than 15 nm exist, which, however, are not detectable by N2 adsorption. The authors should try to perform Hg intrusion in addition or at least must mention this fact.

5. Table 1 and Fig. 5: Related to the by a factor of 2-3 higher pore volume and surface area of the nanoribbons and nanoparticles in comparison to P25, the higher activity of NR and NP is not surprising. Moreover, corrected to a normalized surface area the degradation activity of P25 appears to be even higher. Thus, the concluded high activity is misleading. It is necessary to compare the NR and NP samples to a commercial high-surface TiO2 photocatalyst, like for example Hombikat 100.
Moreover, degradation rates must be calculated and discussed.

6. Conclusions, line 312: I do not see any control over size, shape and morphology of the formed anatase particles be described in the paper. If you have such control, please explain, how you can shift the synthesis route to either the (more or less) exclusive formation of nanoribbons (NR) or to the exclusive formation of nanoparticles (NP). It would be also interesting how distinct rations of NR:NP can be obtained and how reproducible this synthesis route is.

English must be improved

Author Response

Dear reviewer,

Firstly, we would like to thank the referees and editors for their time and efforts in the consideration of this manuscript. We have read and considered their responses and believe we have addressed their concerns point-by-point as below:

Comments and Suggestions for Authors

The paper reports on photocatalytically highly active hierarchical TiO2 nanoribbons/nanosheets prepared solvothermally. As a test reaction only the – well-known, but also very unspecific – methylene blue degradation is used.

Many of the claims given in the paper are not well-proven and documented. Moreover, there are a lot of spelling and grammar errors in the text.

In total, I cannot recommend publication of the paper, since a number of facts are not well presented – and especially scientific explanations are missing.

Nevertheless, the paper is widely well-written and can be published. However, the authors should consider the following points in a revision of the paper:

Reply: We are very thankful to the reviewer for his valuable comments and scientific advice that would make our research more suitable for publication in the Nanomaterials journal.

As organic dyes such as Rhodamine B and Methylene blue are widely used to examine the photocatalytic activity of synthesized catalysts, we used the available standard Rhodamine B dye for testing the photodegradation by our synthesized materials.       

  1. Chapter 3.1: The synthesis mechanism remains unclear. Especially I do not understand for what reasons either nanoribbons or nanoparticles form? What initiates the formation of the nanoribbons? Is there any proof for the proposed “regular repeating structures” (line 207)? How can the ratio of nanoribbons to nanoparticles formed be controlled? The last sentence of the paragraph claims that by-products (line 211) play a role – what kind of by-products are these and do the authors have any proof for their formation?

Reply 1: Further literature research works about the synthesis of hierarchical TiO2 nanostructures with/without using surfactants and their advantages/disadvantages have been added to the introduction. More references and explanations have been added to the synthesis mechanism section. At the beginning of the synthesis mechanism section, we explained the effect of solvents either ethanol or acetic acid on the TiO2 phase formation and control of the morphology. We mentioned that (N.B., the initial stages of formation of amorphous Ti chains were explained previously by our group to synthesize anatase single crystal TiO2 using similar titanium alkoxide (titanium butoxide) with acetic acid) [please see reference [51]. These products in this work, are part of a series of anatase TiO2 morphologies, that we synthesized using a facile one-step solvothermal reaction without using any templates or surfactants. We synthesized different morphologies by controlling the reaction using ethanol and acetic acid.  Some of these series morphologies have already been published and we would like to keep the others including the initial/intermediate synthesis steps for future publications. [Please see references, [46-48]

  1. In the paragraph starting below Fig. 2, i.e., lines 234 – 246 the authors write several times “Fig. 6”, but they mean “Fig. 3”.

Reply 2: the figure numbers in the text have been edited accordingly.

  1. Line 240: Where are the “700 nanoribbons…” coming from? This sentence is not clear to me – please explain with more details.

In line 264 it is said that the nanoribbons might be 700 nm in size – what might define that size, i.e., what limits the interaction of the nanospindles and nanosheets to that size? The mechanism of the aggregation remains fully unclear.

Reply 3: 700 nm was written by mistake; however, we didn’t mean by 700 nm mentioned elsewhere in the text, is the size of nanoribbons, we approximately estimated the size of the void made by the micro-size of TiO2-NR-like spherical shape (around 700 nm, please see SEM and TEM images). The sentence has been edited.

Most of BET equipment including ours has a limit to measure such sub-micro size voids over 500 nm.

 We edited this sentence regarding the estimation values of voids or pores among the TiO2-NR hierarchical structures to be more understood.

  1. Fig. 4: Since at p/p0 close to 1 the adsorption is still increasing strongly, it must be assumed that also pores with diameters larger than 15 nm exist, which, however, are not detectable by N2 adsorption. The authors should try to perform Hg intrusion in addition or at least must mention this fact.

Reply 4:  We measured the pore size distributions for materials by Barrett, Joyner, and Halenda (BHJ) (inset in Figure 4 a), which is combined with the Brunauer-Emmet-Teller (BET) surface area measurements. We mentioned performing Hg intrusion in addition to N2 adsorption/desorption isotherms to detect larger pore sizes, as suggested.

  1. Table 1 and Fig. 5: Related to the by a factor of 2-3 higher pore volume and surface area of the nanoribbons and nanoparticles in comparison to P25, the higher activity of NR and NP is not surprising. Moreover, corrected to a normalized surface area the degradation activity of P25 appears to be even higher. Thus, the concluded high activity is misleading. It is necessary to compare the NR and NP samples to a commercial high-surface TiO2 photocatalyst, like for example Hombikat 100.

Moreover, degradation rates must be calculated and discussed.

Reply 5: We compared our synthesized product with available commercial P25 which is widely used as a standard material for comparing and measuring the photocatalytic activity.

No, we disagree with this claim, (Moreover, corrected to a normalized surface area the degradation activity of P25 appears to be even higher),. We added the BET data to Table 1 and Figure 5 a, and clearly showed the higher photocatalytic/photocurrent activities of our synthesized materials TiO2-NR and TiO2-NP compared to P25. We cannot understand this sentence’’ Moreover, corrected to a normalized surface area the degradation activity of P25 appears to be even higher. Thus, the concluded high activity is misleading).

  1. Conclusions, line 312: I do not see any control over the size, shape, and morphology of the formed anatase particles described in the paper. If you have such control, please explain, how you can shift the synthesis route to either the (more or less) exclusive formation of nanoribbons (NR) or to the exclusive formation of nanoparticles (NP). It would be also interesting how distinct rations of NR: NP can be obtained and how reproducible this synthesis route is.

Reply 6: These products in this work are part of a series of anatase TiO2 morphologies that we synthesized using a facile one-step solvothermal reaction without using any templates or surfactants. We synthesized different morphologies by controlling the reaction using ethanol and acetic acid.  Some of these series morphologies have already been published and we would like to keep the others including the initial/intermediate synthesis steps for future publications. [Please see references, [46-48].

Best regards,

Authors,

Reviewer 4 Report

The article is devoted to the synthesis and characterization of nanoparticles based on TiO2, which are effective for a number of applications. Hierarchically aggregates of anatase TiO2 nanoribbons/nanosheets (TiO2-NR) and anatase TiO2 nanoparticles (TiO2-NP) were produced via one-step solvothermal synthesis using acetic acid or ethanol and titanium isopropoxide as solvothermal reaction systems. The crystalline structure, crystalline phase, and internal and external morphologies of synthesized materials were characterized by several techniques. The goal of this study, was to synthesize the hierarchical structure of anatase TiO2 nanoribbons/nanosheets and anatase TiO2 nanoparticles by solvothermal method and compared their photocatalytic activity and photocurrent with commercial (P25) TiO2. As it was expected, the properties of the synthesized nanoparticles surpassed the standard commercial TiO2 (P25). The most interesting is the fact thatTiO2-NP prepared by the same method showed lower photocatalytic/photocurrent activities than TiO2-NR. The authors explain it by the reduced surface area of TiO2-NP and increased recombination losses due to the spherical shape and smaller size of the nanoparticles. It follows from the article that the superior photocatalytic performance of TiO2-NR was largely due to a special network structure in which individual nanoparticles are combined and which is visible in fig. 2 and 3 (c-f). It should be emphasized that this result was achieved experimentally by a simple method of "soft" and well controlled solvothermal synthesis. However, for a completer and more reasonable suggesting their potential for various applications, such as environmental remediation, solar energy conversion, water splitting, air purification, and renewable energy production it is not enough to perform one test reaction such as organic Rhodamine B. There are no comments on the results obtained. They appear to be credible and reasonable. Authors should pay more attention to the preparation of the text of the article for publication.

Comments and Suggestions for Authors.

These remarks must be corrected!

1.     P.1, line 20. The letter "l" is missing from the word Sovothermal.

2.     P.1, line 21. TiO2 nanorib-20 bion/Nanosheets; The letter highlighted in red must be lowercase.

3.     P3, line 73. TiO2 Nanostructures The letter highlighted in red must be in lowercase.

4.     P.2, lines 76-77 and 87-88 are almost the same. This part of the text should be edited.

5.     P.3, line 118. The Precursor/ The letter highlighted in red must be lowercase.

6.     P.4, line 196. TiO2-NR and TiO2-NR. Probably, different particles are meant, and this should be corrected.

7.     P.5, line 206. to form chain Ti chains. One of the words highlighted in red is redundant.

8.     P.6, lines 236, 243. Figures 6 a-f., Figure 6a-b, Figures 6b, e, Number of these figures. Is 3. See P.7

9.     P.8, line 274. While the 273 average pore size was observed comparable. This is an unfinished sentense that needs editing.

10.  P.8, line 281. of TiO2-NR, TiO2-NP, and materials. The sentence is not finished, probably P25 was ment. See Table 1.

11.  P.9, line303. FOR P25 The word marked in red should be replaced by lowercase.

12.  P. 9, line 306. Figure 5. (a-c) photocatlytic and, (d) photocurrent measurements. The first parenthesis should not contain a dash, but a comma. In the second bracket first letter should be b.

The article contains new interesting results, but it is written very carelessly, not checked by the authors, which makes the reviewing difficult.

Author Response

Dear reviewer,

Firstly, we would like to thank the referees and editors for their time and efforts in the consideration of this manuscript. We have read and considered their responses and believe we have addressed their concerns point-by-point as below:

Comments and Suggestions for Authors

The article is devoted to the synthesis and characterization of nanoparticles based on TiO2, which are effective for a number of applications. Hierarchically aggregates of anatase TiO2 nanoribbons/nanosheets (TiO2-NR) and anatase TiO2 nanoparticles (TiO2-NP) were produced via one-step solvothermal synthesis using acetic acid or ethanol and titanium isopropoxide as solvothermal reaction systems. The crystalline structure, crystalline phase, and internal and external morphologies of synthesized materials were characterized by several techniques. The goal of this study was to synthesize the hierarchical structure of anatase TiO2 nanoribbons/nanosheets and anatase TiO2 nanoparticles by solvothermal method and compared their photocatalytic activity and photocurrent with commercial (P25) TiO2. As it was expected, the properties of the synthesized nanoparticles surpassed the standard commercial TiO2 (P25). The most interesting is the fact that TiO2-NP prepared by the same method showed lower photocatalytic/photocurrent activities than TiO2-NR. The authors explain it by the reduced surface area of TiO2-NP and increased recombination losses due to the spherical shape and smaller size of the nanoparticles. It follows from the article that the superior photocatalytic performance of TiO2-NR was largely due to a special network structure in which individual nanoparticles are combined and which is visible in fig. 2 and 3 (c-f). It should be emphasized that this result was achieved experimentally by a simple method of "soft" and well-controlled solvothermal synthesis. However, for a completer and more reasonable suggestion their potential for various applications, such as environmental remediation, solar energy conversion, water splitting, air purification, and renewable energy production it is not enough to perform one test reaction such as organic Rhodamine B. There are no comments on the results obtained. They appear to be credible and reasonable. Authors should pay more attention to the preparation of the text of the article for publication.

  • The controlling of TiO2-NR nanostructure and morphology was emphasized and described in detail in the synthesis mechanism section, additionally, the suggested phrase has been added to this section.
  • We mentioned in the abstract, the introduction and conclusion to the general potential applications of these synthesized materials based on the principal measurements of photocatalytic/photocurrent in terms of the charge separation and charge transport respectively. These synthesized materials may be considered for further potential applications by us or other researchers in the future. The potential application phrases have been edited accordingly in the abstract and conclusion sections.

Comments and Suggestions for Authors.

These remarks must be corrected!  

  1. P.1, line 20. The letter "l" is missing from the word Sovothermal. Corrected
  2. P.1, line 21. TiO2 nanoribbons/Nanosheets; The letter highlighted in red must be lowercase. Corrected
  3. P3, line 73. TiO2 Nanostructures The letter highlighted in red must be in lowercase. Corrected
  4. P.2, lines 76-77 and 87-88 are almost the same. This part of the text should be edited. Edited
  5. P.3, line 118. The Precursor/ The letter highlighted in red must be lowercase. Corrected
  6. P.4, line 196. TiO2-NRand TiO2-NR. Probably, different particles are meant, and this should be corrected. Corrected
  7. P.5, line 206. to form chain Ti chains. One of the words highlighted in red is redundant. Corrected
  8. P.6, lines 236, 243. Figures 6 a-f., Figure 6a-b, Figures 6b, e, Number of these figures. Is 3. See P.7 Corrected
  9. P.8, line 274. While the 273 average pore size was observed comparable. This is an unfinished sentence that needs editing. Corrected to be read as’’ While the average pore sizes were observed to be almost similar.
  10. P.8, line 281. of TiO2-NR, TiO2-NP, and materials. The sentence is not finished, probably P25 was meant. See Table 1. Corrected
  11. P.9, line 303. FORP25 The word marked in red should be replaced by lowercase. Corrected
  12. P. 9, line 306. Figure 5. (a-c) photocatalytic and, (d) photocurrent measurements. The first parenthesis should not contain a dash, but a comma. In the second bracket, the first letter should be b. Corrected

The article contains new interesting results, but it is written very carelessly, and not checked by the authors, which makes the reviewing difficult.

Re: We revised the manuscript accordingly.

Best regards,

Authors,

Round 2

Reviewer 3 Report

The paper has been significantly improved, however, there is still a point which the authors should consider more:

It is claimed several times in the text (e.g., also in the Conclusions) that the nanoribbons and nanoparticles were formed without any template or surfactant. The point is, how the authors define a "template"? It is known that also some acids - here acetic acid is unsed as a solvent - can have a "structure-directing effect". For larger organic acids, e.g., benzoic acid,  this is even more pronounced. I suggest that the authors have a look whether the addition of such a discussion might be useful. 

There are still some typing errors in the text. Thus, an extensive and careful proof-reading is essentiell. 

Author Response

Responses to the reviewer 3 comments /Nanomaterials Journal

Firstly, we would like to thank the reviewer for his valuable time and efforts spent in, comments, suggestions, and recommendations for consideration of our manuscript. We have read and considered his comments and believe we have addressed his concerns point-by-point as below:

Comments and Suggestions for Authors

The paper has been significantly improved, however, there is still a point that the authors should consider more:

It is claimed several times in the text (e.g., also in the Conclusions) that the nanoribbons and nanoparticles were formed without any template or surfactant. The point is, how do the authors define a "template"? It is known that also some acids - here acetic acid is used as a solvent - can have a "structure-directing effect". For larger organic acids, e.g., benzoic acid,  this is even more pronounced. I suggest that the authors have a look at whether the addition of such a discussion might be useful.

Reply: We kindly acknowledge that, to the best of our knowledge, acetic acid is not typically considered a surfactant-based solvent for synthesizing nanoparticles. It is primarily used as a solvent for certain types of reactions and as a pH adjuster in nanoparticle synthesis processes. Surfactants, on the other hand, are molecules that have both hydrophilic (water-loving) and hydrophobic (water-repelling) properties. They are commonly used to stabilize nanoparticles and prevent their aggregation during synthesis.

We defined the surfactant and templates and addressed their advantages/disadvantages thoroughly in the introduction. 

We kindly refer to lines 136-140 ‘’Surfactants and templates are essential in directing the growth of TiO2 nanoparticles and facilitating their self-assembly into hierarchical structures. Surfactants, such as Cetrimonium bromide (CTAB) or Polyvinylpyrrolidone (PVP), stabilize the nanoparticles and control their size and shape. Templates provide a framework for hierarchical assembly, resulting in specific morphologies such as nanorods, nanotubes, or mesoporous structures [33].’’

 We also refer to Line 241,  ‘’acetic acid has the ability to promote the hydrolysis and condensation of titanium precursor compounds and can also regulate the acidity or alkalinity of the solution involved in the reaction [46-48].’’

And Lines 263-265, ‘’Finally, calcination results in the formation of TiO2-NR or TiO2-NP, with esters acting as an implicitly established self-template (edited and highlighted in the manuscript) while going through the process of formation [45].’’

We agree on the term stated by the reviewer explaining the acetic acid reaction as a‘’ structure-directing effect’’, it has been added to the proposed synthesis mechanism section. (Edited and highlighted in the manuscript)

Comments on the Quality of English Language

There are still some typing errors in the text. Thus, extensive and careful proofreading is essential.

Reply: Proofreading has been done carefully as requested.

Best wishes,

The corresponding author,